# Hypnotizability and Disordered Personality Styles in Cluster A Personality Disorders

**DOI:** 10.3390/brainsci13020182

**Published:** 2023-01-22

**Authors:** Bingren Zhang, Bing Pan, Jueying Chen, Junjie Wang, Zhenyu Zhu, Timo Juhani Lajunen, Wei Wang

**Affiliations:** 1School of Clinical Medicine, Hangzhou Normal University, Hangzhou 311121, China; 2Center for Cognition and Brain Disorders, The Affiliated Hospital, Hangzhou Normal University, Hangzhou 310015, China; 3Department of Psychiatry, Second Affiliated Hospital, School of Medicine, Zhejiang University, Hangzhou 310009, China; 4Department of Psychology, Norwegian University of Science and Technology, 7034 Trondheim, Norway

**Keywords:** cluster A personality disorders, hypnotizability, paranoid, schizoid, schizotypal

## Abstract

Aim: Interpersonal sensitivity and mistrust are the main characteristics of cluster A personality disorders (CAPD) which might be due to the high accessibility to negative suggestions from environments. Yet the exact associations between hypnotic suggestibility and their personality disorder functioning styles remain unclear. Methods: We invited 36 patients with CAPD and 115 healthy volunteers to undergo the Stanford Hypnotic Susceptibility Scale: Form C (SHSS:C) and Parker Personality Measure (PERM). Results: Compared to controls; patients scored higher on PERM paranoid; schizoid; schizotypal; borderline; avoidant; and dependent styles; on the SHSS:C total and “challenge suggestions”, and the passing rates of “hand lowering”, “arm rigidity”, “dream”, and “arm immobilization”. In patients, “dream” negatively predicted the schizoid; “hallucinated voice” negatively the schizotypal; “mosquito hallucination” positively the histrionic and dependent; and “arm immobilization” negatively the avoidant style. Conclusions: Our results suggested that the insusceptibility to perceptual suggestions from others and the high control over body contribute to the paranoid attitude and interpersonal avoidance in CAPD. These findings help to understand the cause of interpersonal problems in these patients and suggest the trial of hypnotherapy for them.

## 1. Introduction

Cluster A personality disorders (CAPD, including the paranoid, schizoid and schizotypal types) are characterized by “odd and eccentric” behaviors and share some similar clinical symptomatology with schizophrenia [1], but both their pharmaco- and psychotherapeutic treatment effects are far from satisfactory [2,3,4]. Reasons for their poor therapeutic effects are diverse, and one of them is due to insufficient research regarding their pathology. A twin study has shown that CAPD is largely determined by genes, and by environmental risk factors as well [5]. In clinics, limited evidence has shown that the paranoid personality disorder is marked heavily by suspicious- and hostile-related perceptual-cognitive “positive” symptoms, the schizoid by extreme social isolation stemming from a lack of desire for interpersonal relationships, and the schizotypal by both perceptual-cognitive “positive” deficits and interpersonal problems [2]. In general, CAPD patients display interpersonal mistrust and sensitivity, even those related to doctors and therapists [3,4]. The interpersonal problems as the core symptoms in these patients were reported to be highly correlated with the negative core beliefs about the self and others [6,7], and the perception of dissimilarity of the self with others [6,8]. Notably, evidence has shown that negative self-concept is closely related to the high accessibility to negative suggestion from nearby people such as parents or peers [9,10], and the negative trait underlies the cognitive failures and schizotypy [11]. Meanwhile, the accessibility to environmental or interpersonal clues of suggestions is associated with the hypnotizability, which is the specific ability to experience the physiological, sensational, emotional, cognitive, or behavioral changes suggested by a hypnotist [12]. The ability is relatively stable after adulthood without training experience [13]. Therefore, one might ask whether hypnotizability contributes to the disordered personality in CAPD with features of interpersonal sensitivity.

Some studies have suggested that hypnotizability could be triggered by intense emotional experience in the trauma-related disorders, somatization disorder, or dissociative identity disorder [14]. Meanwhile, CAPD patients were reported to experience more childhood maltreatment and trauma [15], which even led to suicidal ideation and behavior in schizotypal personality disorder patients [16], suggesting that CAPD patients might possess a high hypnotizability level. Moreover, dissociation of sensorimotor areas happened during hypnosis [17,18]. In high hypnotizable ones, a reduction of glutamate and an increment of GABA in the anterior cingulate cortex [19], a weakened connectivity between the executive control network and the default mode network [20], and a reduced metacognition of agency [21] were also detected. Interestingly, the dysfunction of the frontotemporal cortical network [22] and the metabolic decrement of glutamate in the prefrontal area were found [23], and lower sensorimotor gating with greater volume in the motor pathways were reported in patients with schizotypal traits [24,25]. These findings implied that their top-down awareness of intention was inhibited, and their executive control system would easily lose its control over the sensorimotor domain [17,18,21], which might also lead to a high passing rate of hypnotic suggestions. In addition, the perceptual and cognitive distortions in CAPD patients [26] suggest that they experience similar changes under hypnotic perceptual-cognitive suggestions. On the other hand, the higher paranoia in the low hypnotizable individuals [27] might suggest that paranoid and schizotypal personality disorder patients have more difficulty in passing the perceptual-cognitive suggestions than motor items. The tendency towards social detachment and emotional restriction might also prevent the schizoid personality disorder patients from being deeply suggested by a hypnotizer [28]. By and large, there might be a relatively high hypnotizability level in general, especially for the motor suggestions rather than the perceptual-cognitive ones in CAPD.

Interestingly, abundant evidence showed an association between hypnotizability and personality traits, such as fantasy proneness, absorption [29], extraversion [30], openness to experience [31], and self-transcendence [32]. Thus, the results imply a predictability of personality disorder functioning styles by the alterations of cognition or behavior under hypnosis in CAPD. Other supporting evidence is that the passing rate of posthypnotic amnesia was predictive of the schizoid personality disorder functioning style in general personality disorder patients [33], although the hypnotic manifestations in their patients were not differentiated in precise personality disorder clusters. Recently, Kaczmarska et al. [27] found that compared with medium and high hypnotizable individuals, the low hypnotizable individuals had higher deviation and paranoia scores. This outcome might imply a counterpart between low suggestibility to perceptual-cognitive items and more obvious personality disorder functioning styles in CAPD.

Consistent with previous reports that CAPD patients, especially older adults, seldom ask for help from hospital [3,4]. The CAPD patients recruited in the current study were young adults asking for psychological counseling in a university campus-based clinical setting. As a control group for balanced age and sex distribution, the health volunteers were recruited from university students. Participants underwent the Stanford Hypnotic Susceptibility Scale: Form C (SHSS:C) [34] and the Parker Personality Measure (PERM) [35] for the measurement of their hypnotizability and personality disorder functioning styles. Based on the above-mentioned rationales, we have hypothesized that (1) compared with healthy volunteers, CAPD patients have higher SHSS:C total scores and passing rates of motor suggestions; (2) performance under perceptual-cognitive suggestions negatively predicts the paranoid, schizoid, and schizotypal personality disorder styles in CAPD patients.

## 2. Methods

### 2.1. Participants

We enrolled 36 patients with cluster A personality disorders (CAPD; 19 men and 17 women; aged 20.14 years ± 1.27 S.D., ranged 18~23 years; 4 with paranoid, 28 schizoid, 2 schizotypal, and 2 both paranoid and schizotypal types), and 115 healthy university students as healthy volunteers (controls; 52 men and 63 women; aged 20.42 ± 1.03, ranged 17~22). Patients were received by a psychological counselor (BZ), then diagnosed according to DSM-5 criteria [1] by two psychiatrists (WW and BP), and the diagnoses were later confirmed by the personality measure (PERM, see later). Patients were with at least one T score no less than 60 on PERM paranoid, schizoid, and schizotypal styles, and they should not be comorbid with cluster B or C personality disorders according to DSM-5 criteria. Most individuals with CAPD also reported anxiety, depression, or sleep problems, but they were all free of psychiatric disorders such as dissociative identity disorder, trauma related disorders, phobias, and schizophrenia. A semi-structured interview was performed with each healthy participant to exclude psychiatric or neurological problems. All participants were right-handers, had no prior hypnotic experience (which might influence their hypnotizability level [36]), had slept for more than 6 h the previous night, and were free of drugs and alcohol for at least 72 h prior to the tests.

### 2.2. Measures

#### 2.2.1. The Stanford Hypnotic Susceptibility Scale: Form C

SHSS:C [34] is a commonly used 12-item standard test that measures response to hypnotic induction suggestions. It includes two direct suggestions for motor performance (ideomotor suggestions), two suggestions for loss of arbitrary motor control (challenge suggestions), and eight suggestions for changes in individual perception, memory, or cognition (perceptual-cognitive suggestions). This scale has been shown to be reliable [37] and valid [38] in many countries, including China [39].

The individual SHSS:C test was administered by one of the authors who was blind to the group of participants. Each suggestion successfully made (as judged by the hypnotizer according to objective criteria of SHSS:C) was counted as a point. The percentage of participants who had passed the item was defined as the “passing rate”. The hypnotizability of participants was divided into three levels from low (including very low) to high (including very high, Table 1) [34].

#### 2.2.2. The Parker Personality Measure (PERM)

PERM [35] is a 92-item self-assessment scale for 11 functioning styles of personality disorder: (1) paranoid, (2) schizoid, (3) schizotypal, (4) antisocial, (5) borderline, (6) histrionic, (7) narcissistic, (8) avoidant, (9) dependent, (10) obsessive-compulsive, and (11) passive-aggressive style. Each PERM item consists of a 5-point Likert scale (1—very unlike me, 2—moderately unlike me, 3—somewhat unlike and like me, 4—moderately like me, 5—very like me). The Chinese version of PERM has already proven to be reliable in China [40].

### 2.3. Statistical Analysis

SPSS IBM Statistics v.20.0 was used for data analyses. The distributions of gender, education, and experience of hypnosis in two groups were analyzed by the Mann–Whitney U test. The Shapiro–Wilk test was used to verify the normal distribution of the total SHSS:C score. The age and SHSS:C scores in two groups were dealt with the Student’s *t*-test. Since each SHSS:C item measures a different aspect of hypnotizability [28], the passing rate of each item was compared by the Mann–Whitney U test. Afterwards, multivariate ANOVA plus post-hoc Bonferroni test were used to see the main effect of and interaction effect between group (HC/CAPD) and hypnotizability level (low/medium/high) on PERM styles (with T scores). The Pearson Correlation Analysis and the Multiple Linear Regression Analysis (Stepwise Method) were used successively to explore the relationships between SHSS:C and PERM in two groups. For predictions, demographic variables and the passing rates of SHSS:C items were taken as potential predictors. The alpha value (*p*) was set to 0.05.

## 3. Results

No significant difference was found between the two groups regarding age (t = 1.33, df = 149, *p* = 0.185), gender (U = 1913.50, *p* = 0.429), education level (U = 1960.00, *p* = 0.084), or hypnosis experience (U = 1948.00, *p* = 0.216).

The SHSS:C internal reliability was 0.70 in the current study. The distribution of SHSS:C experience in the current sample (N = 151) was however abnormal (W = 0.97, *p* = 0.002), as previously reported [31,32]. The total scores on SHSS:C (controls: 5.55 ± 2.54; CAPD, 6.61 ± 2.80; t = 2.14, *p* = 0.034, Cohen’s d = 0.38), on the challenge suggestions (controls: 1.03 ± 0.86; CAPD, 1.44 ± 0.69; t = 2.61, *p* = 0.010, Cohen’s d = 0.48), and on the passing rates of “hand lowering” (OR = 0.33), “arm rigidity” (OR = 0.37), “dream” (OR = 0.33), and “arm immobilization” (OR = 0.44) items were higher in patients than in controls (Table 1). No other significant difference was found on SHSS:C between groups.

The PERM internal reliability in the current sample was 0.90. The main (group) effect was found on the paranoid (partial η^2^ = 0.03), schizoid (partial η^2^ = 0.46), schizotypal (partial η^2^ = 0.24), borderline (partial η^2^ = 0.08), avoidant (partial η^2^ = 0.08), and dependent (partial η^2^ = 0.04) styles (F = 4.28~124.00, MSE = 232.23~6490.83, *p* = 0.000~0.040), with those styles higher in patients (Table 2). No main effect of hypnotizability or no significant interaction effect between group and hypnotizability was found on PERM styles.

There were associations between the performance on SHSS:C and PERM styles in the two groups (see Table 3 for correlation and Table 4 for prediction results). To reduce false positives, only correlations with *p* less than 0.01 were treated as meaningful. In healthy volunteers, the passing rate of “arm immobilization” (*p* = 0.009) positively correlated with the histrionic style, “age regression” positively predicted the schizoid (*p* = 0.043) and obsessive-compulsive (*p* = 0.024) styles, “arm immobilization” (*p* = 0.003) and “age regression” (*p* = 0.012) positively while “anosmia to ammonia” (*p* = 0.002) negatively predicted the histrionic style. In patients, the passing rate of “hallucinated voice” negatively correlated with (*p* = 0.006) and predicted (*p* = 0.006) the schizotypal style, “dream” negatively predicted the schizoid style (*p* = 0.013), “mosquito hallucination” positively predicted the histrionic style (*p* = 0.035), arm immobilization (*p* = 0.021) negatively predicted the avoidant style, “mosquito hallucination” (*p* = 0.016) positively predicted the dependent style.

## 4. Discussion

In this study, we found higher SHSS:C total score in CAPD patients, demonstrating that they were more easily hypnotized, as reported in schizophrenia [41], supporting the cluster A and schizophrenia disorder spectrum [1,14]. On the one hand, the high suggestibility in patients accounts for their interpersonal sensitivity when combined with the previous report of their childhood maltreatment and trauma experience [15] and implies that hypnotherapy is a therapeutic choice for these patients as it is for schizophrenia [42]. The higher passing rates of challenge suggestions such as “arm rigidity” and “arm immobilization” in patients have suggested that their functional connectivity between the salience network and the executive control network were more strengthened, which ensured bottom-up sensorimotor control [20]; however, their executive control system was further weakened, and they easily lost their top-down arbitrary motor control [43]. A similar pattern of motor control was reported in individuals with schizotypal traits, regarding the impaired prepulse inhibition [25]. The higher passing rate of “dream” in CAPD was consistent with a previous report on general personality disorders [33], suggesting that the metacognition of these patients was more dysfunctional so they could experience alterations in the state of consciousness after dissociation [21,44]. Our results also helped explain the “positive” symptoms in CAPD similarly found in schizophrenia [2]. Altogether, these results support our first hypothesis, and are open to further speculation that the prefrontal cortex function was altered in cluster A patients.

We also found that PERM paranoid, schizoid, schizotypal, borderline, avoidant, and dependent styles were more elevated in patients than in healthy volunteers. In addition to the odd and eccentric features, we have shown that these patients were also more emotional, anxious, and fearful than healthy individuals [2]. Our results were partly in line with a previous report on personality disorder patients in general [33]. In clinics, paranoid schizophrenia also displays the elevated avoidant trait which is considered as the premorbid personality disorder [45]. Our results have thus revealed the weak self in cluster A patients, which contributed to their high interpersonal sensitivity.

Concerning the predictions in healthy volunteers, “age regression” positively predicted the PERM schizoid and obsessive-compulsive styles. “Age regression” is a cognitive-perceptual suggestion which induces people to go back and re-experience their earlier periods of life, and only the upper 10–15% high hypnotizable individuals have passed this item [46]. The predictions on two PERM styles indicated that even in healthy people, high accessibility to unpleasant memories in childhood contributes to the maladaptation in current life, including a lowed desire in interpersonal relationships, and a sense of losing control and compulsive tendency, which is in line with previous report on individuals with trauma experience [47]. The positive correlation between “arm immobilization” and histrionic style as well as the positive predictions of “age regression” and “arm immobilization” on histrionic style in healthy volunteers are supported by the social role-taking theory [48] which proposes the correlation between subjective identification, active attempts to achieve social role construction, and the feeling “like being hypnotized”. Moreover, “anosmia to ammonia” negatively predicted the histrionic style in our healthy volunteers which might be because the intense smell of ammonia causes overreaction in people with histrionic traits.

In cluster A patients, “dream” negatively predicted the schizoid style. Since “dream” involves alterations in the state of consciousness in the presence of a hypnotizer where the participant has to be relaxed in body and mind [44], it is understandable that patients who pass this item tend to be less socially detached and emotionally restricted [28]. The negative intercorrelation and the prediction of passing rate of “hallucinated voice” on the schizotypal style in CAPD patients were partly supported by evidence that individuals who easily followed the hallucination-like suggestion had high openness to experience [31], and high paranoia scores as in the low hypnotizables [27]. These results are in line with our second hypothesis and suggest that the insusceptibility to perceptual suggestions from other persons could aggravate the paranoid attitude and interpersonal isolation, which were in accordance with the high drop off rate from therapy and the poor therapeutic effects [2,3,4]. We also found the passing rate of “mosquito hallucination” positively predicted the histrionic and dependent styles in CAPD patients, suggesting that the easy experience of positive perception contributes to the tendency to seek attention from and rely on others on the one hand [2], and is in line with their deficiency and intrinsic need for attachment security on the other [49]. Furthermore, the lower passing rate of “arm immobilization” was associated with the higher avoidant style, which might imply that the stronger their tendency to use body-controlling as self-protection, the lower risk they take to devote to interpersonal relationships.

Furthermore, correlations between some demographic variables and personality disorder functioning styles were found. The positive prediction of the female gender for borderline style in healthy people was consistent with the previous demographic reports on borderline personality disorder in clinics [1]. In addition, the positive prediction of the female gender for avoidant and dependent styles in CAPD might indicate that female patients were more inclined to be sensitive, by avoiding interpersonal communications as well as relying on someone they trust [35]. The negative prediction of age for dependent style in CAPD might indicate that young age exacerbates the low self-evaluation and obedience in these patients [35].

One should be aware of the limitations of our study. Firstly, participants recruited in our study were young adults or young university students; whether our results could be generalized to other ages or other international clinical settings need to be clarified. Secondly, most of our CAPD patients were the schizoid type; future studies might enroll more patients suffering from the paranoid and schizotypal types since the three subtypes are heterogeneous. Thirdly, we used a measure of behavioral suggestibility with the passing rate of items which is less sensitive to detect brain activity under hypnosis [50]; future studies might employ cerebral function techniques to look for these possible changes. Nevertheless, we found that CAPD patients (mainly the schizoid type) were easier to hypnotize, by the loss of casual movements and experiencing the change of consciousness. In these patients, the insusceptibility to perceptual suggestions from others and the control over their body are linked with their paranoid attitude and interpersonal avoidance, and their elevated suggestibility is associated with attention seeking and dependence on others. Our findings encouraged a clinical trial of hypnotherapy in cluster A personality disorders.

## Figures and Tables

**Table 1 brainsci-13-00182-t001:** The distributions of participants who passed the hypnotic susceptibility tests in healthy volunteers (controls, *n* = 115) and cluster A personality disorders (CAPD, *n* = 36).

	Controls		CAPD		U	*p*
	*n*	Passing Rate	*n*	Passing Rate		
Low hypnotizability (passed 0–3 items)	23	20.0%	11	30.6%	-	-
Medium (4–8)	73	63.5%	20	55.6%	-	-
High (9–12)	19	16.5%	5	13.9%	-	-
01 Hand lowering	77	67.0%	31	86.1%	1673.50	**0.027**
02 Moving hands apart	79	68.7%	28	77.8%	1882.00	0.297
03 Mosquito hallucination	69	60.0%	19	52.8%	1920.50	0.445
04 Taste hallucination	73	63.5%	21	58.3%	1963.50	0.580
05 Arm rigidity	65	56.5%	28	77.8%	1630.00	**0.023**
06 Dream	36	31.3%	21	58.3%	1510.50	**0.004**
07 Age regression	88	76.5%	27	75.0%	2038.50	0.852
08 Arm immobilization	54	47.0%	24	66.7%	1662.00	**0.040**
09 Anosmia to ammonia	29	25.2%	10	27.8%	2017.00	0.760
10 Hallucinated voice	10	8.7%	4	11.1%	2020.00	0.664
11 Negative visual hallucination	41	35.7%	15	41.7%	1945.50	0.516
12 Posthypnotic amnesia	17	14.8%	10	27.8%	1801.00	0.077

Note: U, the Mann–Whitney U test; *p* < 0.05 were in bold.

**Table 2 brainsci-13-00182-t002:** Mean raw and T scores (mean ± S.D) of the Parker Personality Measure in healthy volunteers (controls, *n* = 115) and cluster A personality disorders (CAPD, *n* = 36).

	Controls		CAPD	
	Raw Score	T Score	Raw Score	T Score
Paranoid	20.88 ± 5.26	40.41 ± 6.22	25.72 ± 9.05	44.78 ± 10.10 *
Schizoid	19.38 ± 2.90	45.77 ± 5.80	27.78 ± 5.32	63.11 ± 10.51 ***
Schizotypal	9.50 ± 2.88	42.25 ± 5.46	14.86 ± 4.00	51.61 ± 7.63 ***
Antisocial	19.50 ± 4.12	43.17 ± 5.03	23.03 ± 5.80	45.58 ± 7.00
Borderline	19.42 ± 5.17	36.57 ± 6.00	25.28 ± 6.70	42.72 ± 8.14 **
Histrionic	12.70 ± 2.77	42.58 ± 6.49	12.56 ± 3.64	41.47 ± 8.96
Narcissistic	17.10 ± 3.76	42.93 ± 5.45	16.94 ± 4.95	42.17 ± 7.35
Avoidant	24.03 ± 5.25	40.04 ± 6.46	28.78 ± 6.16	45.89 ± 7.23 ***
Dependent	22.01 ± 4.67	41.50 ± 5.98	23.61 ± 6.26	43.97 ± 7.72 *
Obsessive-Compulsive	16.32 ± 3.58	43.64 ± 7.16	15.67 ± 4.13	42.86 ± 8.14
Passive-Aggressive	20.06 ± 4.16	40.41 ± 6.22	21.28 ± 5.05	44.78 ± 10.10

Note: * *p* < 0.05, ** *p* < 0.01, *** *p* < 0.001 vs. HC.

**Table 3 brainsci-13-00182-t003:** Correlations between performance on the Stanford Hypnotic Susceptibility Scale: Form C and personality disorder functioning styles of the Parker Personality Measure in healthy volunteers (controls, *n* = 115) and cluster A personality disorders (CAPD, *n* = 36).

	Controls			CAPD			
	Schizoid	Histrionic	Obsessive-Compulsive	Schizoid	Schizotypal	Histrionic	Avoidant
Mosquito hallucination	-	-	-	-	-	0.35 *	-
Taste hallucination	-	-	-	-	-	0.35 *	-
Dream	-	-	-	−0.34 *	-	-	-
Age regression	0.23 *	0.23 *	0.21 *	−0.27	-	-	-
Arm immobilization	-	**0.24 ****	-	-	-	-	−0.41 *
Anosmia to ammonia	-	−0.23 *	-	-	-	-	−0.36 *
Hallucinated voice	-	-	-	-	**−0.45 ****	-	-
Negative visual hallucination	-	-	-	-	−0.35 *	-	-
Score on challenge suggestions	-	-	-	-	-	-	−0.38 *

Note: Only correlations with * *p* < 0.05 were listed, and the correlations with ** *p* < 0.01 were in bold.

**Table 4 brainsci-13-00182-t004:** Predictions of personality disorder functioning styles of the Parker Personality Measure by performance on the Stanford Hypnotic Susceptibility Scale: Form C (gender and age as covariates) using stepwise regression analysis in healthy volunteers (controls, *n* = 115) and cluster A personality disorders (CAPD, *n* = 36).

	a-R^2^	Controls Beta (B, SE), Predictor	a-R^2^	CAPD Beta (B, SE), Predictor
Paranoid	-	-	-	-
Schizoid	0.04	**0.23 (3.13, 1.25), Age regression**	0.09	**−0.34 (−7.12, 3.39), Dream**
Schizotypal	-	-	0.18	**−0.45 (−10.81, 3.66), Hallucinated voice**
Antisocial	0.03	0.19 (2.71, 1.34), Education level	-	-
Borderline	0.08	**0.27 (3.28, 1.09), gender (female)**	-	-
Histrionic	0.16	**0.27 (3.45, 1.13), Arm immobilization** **−0.27 (−4.07, 1.29), Anosmia to ammonia** **0.22 (3.35, 1.32), Age regression**	0.10	**0.35 (6.25, 2.84), Mosquito hallucination**
Narcissistic	0.03	0.19 (2.97, 1.45), Education level	-	-
Avoidant	-	-	0.37	**0.49 (7.05, 1.94), gender (female)** **−0.033 (−4.99, 2.06), Arm immobilization**
Dependent	0.03	0.19 (2.22, 1.11), Gender (female)	0.28	**−0.47 (−2.89, 0.93), Age** **0.39 (6.01, 2.36), Mosquito hallucination** **0.30 (4.64, 2.25), Gender (female)**
Obsessive-Compulsive	0.04	**0.21 (3.55, 1.55), Age regression**	-	-

Note: a-R^2^, adjusted R^2^; only predictors having *p* < 0.05 were listed; predictors with |beta| > 0.20 were bolded.

## Data Availability

The datasets used and/or analyzed during the current study are available from the corresponding or first author on reasonable request.

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
