# Peer review of "Hypnotizability and Disordered Personality Styles in Cluster A Personality Disorders"

_brainsci, 2023, doi:10.3390/brainsci13020182_

Round 1
Reviewer 1 Report
The study is in the same vein of a few attempts to associate hypnotizability with personality disorders. The authors should clarify, on the basis of current evidence, whether they believe that being low or high hypnotizable is a premise to personality disorder. Associations are not always relevant to interpretations. Moreover, similar behaviors in patients and healthy more/less hypnotizable persons do not necessarily indicate similar basis for those behaviors.
The different associations of SHSS itens with scales indicating personality disorders in healthy people and patients should induce to be cautions.
In particular, I am not sure that individual hypnotizability could orient psychotherapy in the studied patients
Author Response
Dear Reviewer#1:
Thank you very much for offering us an opportunity to revise our manuscript. Below we outline in detail the changes we have made according to your comments. Our answers are in these big brackets ‘{}’.
R1.1. The study is in the same vein of a few attempts to associate hypnotizability with personality disorders. The authors should clarify, on the basis of current evidence, whether they believe that being low or high hypnotizable is a premise to personality disorder. Associations are not always relevant to interpretations.
Answer: {Thank our Reviewer #1 for the very thoughtful suggestion. We have put forward our suspicion over this point in the last sentence of Para1 in the Introduction part. We also added more evidence to support our point, such as the studies of childhood maltreatment and trauma experience which might trigger the high hypnotizability in cluster A personality disorders (CAPD), of the neuropsychological and neurochemical characteristics of high hypnotizables which are consistent with those manifested in CAPD patients, and of the association between hypnotizability and personality, etc. (Please see the words in red from Para.s 1 to 3 in the Introduction part for more details)}
R1.2. Moreover, similar behaviors in patients and healthy more/less hypnotizable persons do not necessarily indicate similar basis for those behaviors. The different associations of SHSS items with scales indicating personality disorders in healthy people and patients should induce to be cautions.
Answer: {Yes, we agree with our Reviewer#1 that compared with healthy people, patients who have abnormal personality functioning styles might have different bases for their performance under hypnosis. Therefore, for the different associations of SHSS items with scales indicating personality disorders (i.e., predictions for the Schizoid and Histrionic styles), we added more explanations based on the pathopsychological features and clinical manifestations of patients, please see lines 252-256, 265-267.}
R1.3. In particular, I am not sure that individual hypnotizability could orient psychotherapy in the studied patients
Answer: {We are sorry for the misunderstandable expressions in our last manuscript. We totally agree with our Reviewer#1 that hypnotic susceptibility cannot accurately predict treatment outcomes in patients, although it is an important factor in cognitive hypnotherapy (also refer to Montgomery et al., Int J Clin Exp Hypn, 2011). In the current draft, we have advocated that a clinical trial of hypnotherapy be one of the therapeutic choices for CAPD patients, please see lines 26-27, 213-214, 293-294.}
We hope our changes this time will satisfy you. If we can be of further assistance, please feel free to contact us.
Yours,
Dr. Wei WANG, Department of Psychology, Norwegian University of Science and Technology, Trondheim, Norway
Reviewer 2 Report
Authors explored the relationship between hypnotizability and pathological personality styles. reported below are my comments to the paper:
- line 51: replace "others" with "hypnotist". In general, please, refer to APA definition of hypnotisability, that it, “the ability to experience suggested alterations in physiology, sensations, emotions, thoughts or behavior during hypnosis” (Elkins et al., 2015)
- lines 62-64: in my opinion, this is the most critical point: authors describe hypnotizability as a personality predictor (in fact, a regression analysis is presented). First of all, it is important to say that the here considered hypnotisability is mainly a measure of behavioral suggestibility (see, e.g., Facco 2021 for a critical review) as no phenomenological measures are considered. Moreover, hypnotizability is a modifiable (e.g, Lynn 2004 for a review) and multifactorial construct depending also on relational, cognitive and psychological factors directly associated with personality style, with the latter reflecting more stable traits through lifetime. In other words, my opinion is that the opposite causal relationship could be true (personality as a predictor of hypnotisability). Also, there are previous studies on the personality-hypnotizability relationships, but references on this point are completely missing. See for example Cardeña1 & Terhune 2014 (they also use correlational analysis for exploring these relations..)
Lines 88-90: first hypothesis (patients with higher scores than controls) seem to be in contrast with the study commented above (reference 17) reporting low hypnotisability for patients with higher scores of paranoia. Please, clarify why your hypothesis is the opposite: otherwise, it looks like it was results-guided.
Methods, participants: no information are reported on patients recruitment (they came from doctors’ private practice, hospital, general population….?). This is mandatory both for ethical and methodological reasons: for example, did they look for a therapy or not? This is a fundamental information as it could affect the results interpretation (comments on this point are needed in the discussion).
Lines 116-118: even the SHSS’s authors disputed the “gold standard” conception of these measures. There is an historical debate on this point: please, refer also to limitations of measuring hypnotisability only in terms of “passing rate” of suggestions.
Table: replace number of people with “n” or “subjects”. Total score of SHSS should be reported otherwise (not under the number of people column). “t/U” indices should be described in the table caption
Methods: for the reasons discussed above, I think correlations are more appropriate than regression as no clear inferences can be made on the predictor-dependent variable order (in my opinion, and according to previous studies, personality and hypnotisability are correlated but dissociable constructs).
Discussion: findings are focused on the “interpersonal sensitivity” construct, but it needs to be deepened in the introduction. I think the item-based interpretations are a bit speculative, especially for the reasons discussed above; also, “CAPD” includes very different personality styles, so it is hard to make common conclusions for different psychopathological disorders. This point is a main limitation that need to be reported clearly (also, distribution of the three personality styles is not homogeneous in the sample)
In conclusion, I think the conceptual approach is not correct: if authors agree, correlations are needed, and the paper has to be reviewed accordingly. Otherwise, I think the methodology and findings interpretation are quite critical
Author Response
Dear Reviewer#2:
Thank you for offering us an opportunity to revise our manuscript. Below we outline in detail the changes we have made according to your comments. Our answers are in these big brackets ‘{}’.
R2.1. line 51: replace "others" with "hypnotist". In general, please, refer to APA definition of hypnotisability, that it, “the ability to experience suggested alterations in physiology, sensations, emotions, thoughts or behavior during hypnosis” (Elkins et al., 2015)
Answer: {Thank our Reviewer #2 for the notice. We have replaced the word “others” with “hypnotist”, and we have also modified the definition of hypnotisability according to Elkins et al. (2015). Please see lines 50-51.}
R2.2. lines 62-64: in my opinion, this is the most critical point: authors describe hypnotizability as a personality predictor (in fact, a regression analysis is presented). First of all, it is important to say that the here considered hypnotisability is mainly a measure of behavioral suggestibility (see, e.g., Facco 2021 for a critical review) as no phenomenological measures are considered. Moreover, hypnotizability is a modifiable (e.g, Lynn 2004 for a review) and multifactorial construct depending also on relational, cognitive and psychological factors directly associated with personality style, with the latter reflecting more stable traits through lifetime. In other words, my opinion is that the opposite causal relationship could be true (personality as a predictor of hypnotisability). Also, there are previous studies on the personality-hypnotizability relationships, but references on this point are completely missing. See for example Cardeña1 & Terhune 2014 (they also use correlational analysis for exploring these relations.)
Answer: {Again thank our Reviewer #2 for the very insightful comment. Accordingly, we have added a) previous studies on the personality-hypnotizability relationships to the Introduction part; b) more details about our patients, i.e., had no hypnotic experience before (including training) which could influence their hypnotizability level in the Methods part, and put forward the literature about the stability of hypnotisability after adulthood in general (Piccione et al., 1989); c) the shortcomings of the behavioral measurement in our Limitation part. Please see lines 50-51, 80-84, 119-120, and 285-287 for details.}
R2.3. Lines 88-90: first hypothesis (patients with higher scores than controls) seem to be in contrast with the study commented above (reference 17) reporting low hypnotisability for patients with higher scores of paranoia. Please, clarify why your hypothesis is the opposite: otherwise, it looks like it was results-guided.
Answer: {Thank our Reviewer #2. In the current draft-Introduction, we have added our understanding of the study of Kaczmarska et al. [17] and modified the rationales for and expressions of our hypotheses, please see lines 72-75, 90-91, and 100-101.}
R2.4. Methods, participants: no information are reported on patients recruitment (they came from doctors’ private practice, hospital, general population….?). This is mandatory both for ethical and methodological reasons: for example, did they look for a therapy or not? This is a fundamental information as it could affect the results interpretation (comments on this point are needed in the discussion).
Answer: {Following our Reviewer #2, we have added information about patients’ recruitment in lines 94-95, 110-111, and comments on this point in our Limitations (lines 281-282).}
R2.5. Lines 116-118: even the SHSS’s authors disputed the “gold standard” conception of these measures. There is an historical debate on this point: please, refer also to limitations of measuring hypnotisability only in terms of “passing rate” of suggestions.
Answer: {We are sorry for our imprecise descriptions of SHSS:C, we have deleted the words “gold standard” in the current draft (please see lines 127-131) and modified the Limitation with another point regarding the ability to measure hypnotisability with behavioral tools and “passing rate” indicators and advocated future lab tests of the neural activity under hypnosis. Please also see our limitations (lines 285-287).}
R2.6. Table: replace number of people with “n” or “subjects”. Total score of SHSS should be reported otherwise (not under the number of people column). “t/U” indices should be described in the table caption
Answer: {Following our Reviewer #2, we have modified Table 1, and added the results of the total score on SHSS:C to the Results part, please see Table 1 and lines 168-170.}
R2.7. Methods: for the reasons discussed above, I think correlations are more appropriate than regression as no clear inferences can be made on the predictor-dependent variable order (in my opinion, and according to previous studies, personality and hypnotisability are correlated but dissociable constructs).
Answer: {Thank our Reviewer #2 for the constructive suggestion. We agree with our Reviewer#2 that the rationales for our second hypothesis about the prediction of performance under hypnosis for personality disorder functioning styles in Cluster A need to be strengthened. Therefore, we have reorganized our Introduction part to make our theoretical basis and inference process clearer (please see Para.s 1 to 3 in the Introduction part, and lines 101-103). Meanwhile, we have added the correlation results in the new Table 3 and Results part (please see the new Table 3 and lines 183-195).}
R2.8. Discussion: findings are focused on the “interpersonal sensitivity” construct, but it needs to be deepened in the introduction. I think the item-based interpretations are a bit speculative, especially for the reasons discussed above; also, “CAPD” includes very different personality styles, so it is hard to make common conclusions for different psychopathological disorders. This point is a main limitation that need to be reported clearly (also, distribution of the three personality styles is not homogeneous in the sample)
Answer: {Thank our Reviewer #2 for the insightful notice. In the current draft, we have modified our Para 1 in the Introduction part to describe more the “interpersonal sensitivity” in patients. In our Limitation part, we have added information of the heterogeneity in CAPD patients, please see lines 41-54, and 282-285.}
R2.9. In conclusion, I think the conceptual approach is not correct: if authors agree, correlations are needed, and the paper has to be reviewed accordingly. Otherwise, I think the methodology and findings interpretation are quite critical.
Answer: {Thank our Reviewer #2 for the criticisms which are so constructive. As shown in our answers above (esp. to R2.7.), correlation results have been added, and we modified our Introduction. Methods, Results, and Discussion parts greatly, please see these parts for details.}
We hope our changes this time would satisfy you. If we can be of further assistance, please feel free to contact us.
Yours,
Dr. Wei WANG, Department of Psychology, Norwegian University of Science and Technology, Trondheim, Norway
Round 2
Reviewer 1 Report
The paper has been improved
Reviewer 2 Report
the authors fulfilled my requests